# Unusual Cardiac Manifestations of a Pheochromocytoma in a Girl

**Lisa D'Angelo [1], Anne-Simone Parent [1], Céline Derwael [2], Roland Hustinx [2] and Marie-Christine Seghaye [1,\*]**

1   Department of Pediatrics, CHU of Liège, 4000 Liège, Belgium
2   Division of Nuclear Medicine and Oncological Imaging, CHU of Liège, 4000 Liège, Belgium
\*   Correspondence: mcseghaye@chuliege.be; Tel.: +32-323-9275

**Abstract:** We report the case of an 11-year-old girl who complained about severe asthenia, orthostatic dizziness and abdominal pain for 4 weeks. The primary investigation concluded on febrile urinary tract infection treated by antibiotics. Symptom persistence prompted cardiological and endocrinological investigations. A fluctuation in blood pressure, long QT interval, dilation of the aortic root and left ventricular hypertrophy were documented. Elevated levels of urinary catecholamines together with the presence of a right-sided adrenal mass shown via abdominal ultrasound and magnetic resonance imaging were highly suggestive of a pheochromocytoma. This was confirmed by through iodine-123-metaiodobenzylguathdine ([123I]-mIBG) scintigraphy. Genetic analysis allowed for the exclusion of pathogenic mutations in genes implicated in hereditary paragangliomas and pheochromocytomas but showed a rare somatic mutation in exon 3 of the von Hippel-Lindau gene. The patient was treated with a β-blocker and calcium channel antagonist and underwent laparoscopic right-sided adrenalectomy. Cardiac manifestations resolved soon after surgery indicating that they were secondary to the pheochromocytoma. After 5 years of follow-up, the patient remains asymptomatic without any sign of tumor recurrence. The presence of aortic root dilation, a prolonged QT-interval and left ventricular hypertrophy may be early cardiac manifestations of a pheochromocytoma in a child and should prompt this diagnosis to be evoked.

**Keywords:** aortic root dilation; prolonged QT interval; left ventricular hypertrophy; pheochromocytoma; von Hippel-Lindau gene mutation

## 1. Introduction

Pheochromocytoma is a rare neuroendocrine tumor belonging to the family of neural crest cell-derived neoplasms arising from adreno-medullary chromaffin cells. Pheochromocytoma in children has an incidence of 0.11 per million and is more frequently familial (40%), bilateral (20%) and malignant (10%) compared to that in adults [1,2]. Children with neurofibromatosis, von Hippel-Lindau disease, familial pheochromocytoma/paraganglioma syndrome or multiple endocrine neoplasia syndromes carry a higher risk of developing pheochromocytoma [3]. This tumor excessively produces one or more catecholamines, such as epinephrine, norepinephrine and dopamine. This release and the consecutive increased activation of the sympathetic nervous system are responsible for the symptomatology. Classical symptoms consist of systemic arterial hypertension even with hypertensive crisis, tachycardia, headache and abnormal sweating. Additionally, patients complain about dizziness, weight loss, nausea and vomiting, abdominal and back pain and pale and moist skin. The excessive secretion of catecholamines may be lethal as it leads to organ failure secondary to severe systemic arterial hypertension or hypertensive crisis. In addition, electrophysiological complications, such as a prolonged QT interval and ventricular arrhythmias are described. For all these reasons, rapid diagnosis is mandatory. The diagnosis of pheochromocytoma requires demonstration of the excessive production of cathecholamines. Therefore, blood and urinary levels of epinephrine, norepinephrine and their metabolites,

metanephrine, normetanephrine, dopamine and vanillylmandelic acid, must be determined. Given the variation of catechomaline production during the nychtemeron, catechomanine concentrations in urine are assessed by using a 24-h urine collection.

In cases of proven abnormal catecholamine production, a diagnostic work-up is completed via anatomical imaging with abdominal sonography and magnetic resonance imaging of the kidney and the adrenal lodges. In addition, functional scans, such as iodine-123-metaiodobenzylguathdine ([123I]-mIBG) scintigraphy must be undertaken in order to improve diagnostic specificity and sensitivity and to detect metastases and multifocality.

Surgical treatment of pheochromocytoma via conventional open surgery or laparoscopy provides excellent long-term results in experienced centers. Peri-operative management with treatment by a β-blocker or an α-blocker and calcium channel antagonists is of paramount importance in order to normalize blood pressure and avoid hypertensive crisis and coronary vasospasm that may be elicited by massive catecholamine release during intra-operative tumor manipulation, which can be fatal [4].

## 2. Case Report

We report the case of an 11-year-old girl who had been presenting a severe asthenia for a month, orthostatic dizziness, headaches and abdominal pain. Patient familial history was remarkable due to the total absence of the pericardium associated with hypogammaglobulinemia and bronchiectasis in her 9-year-old sister [5]. Patient personal history was uneventful.

A previous recent laboratory examination had revealed moderate inflammation (leucocyte count: 13,440/mm; CRP: 30 mg/L) but increased sedimentation rate (68 mm/h), and slightly increased fasting glycemia (106 mg/dL) with a normal insulin concentration. Urine analysis showed bacteriuria. On that basis, she received a treatment with sulfamethoxazol.

Patient status did not improve, and thus, she was referred to our institution for endocrinological and cardiological advice. A physical examination confirmed recent weight loss (2.5 kg in 10 days), systemic arterial hypertension (147/104 mm Hg) in an upright position dropping in a supine position (80/60 mm Hg), a normal heart rate and normal transcutaneous oxygen saturation.

The patient was pale with wet and cold extremities. Precordium was hyperactive, and cardiac and pulmonary auscultation was normal as was abdominal palpation.

An electrocardiogram showed left ventricular hypertrophy according to Davignon's criteria [6] with non-specific repolarization anomalies and a prolonged QT interval corrected by the Bazzett's formula (QTc: 470 ms) (Figure 1).

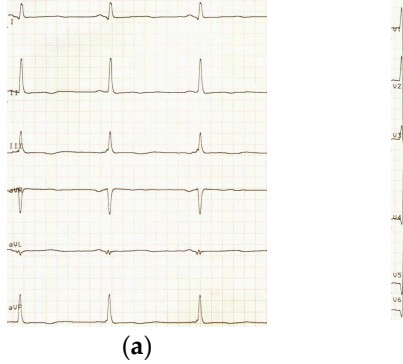 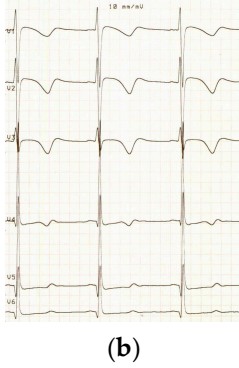

(a)  (b)

**Figure 1.** Twelve-lead electrocardiogram with limb leads (**a**) and chest leads (**b**). QT duration is 430 ms. The calculated duration of the QT interval corrected for heart rate by Bazzett's formula is 470 ms. The recording speed is 50 mm/s and the amplitude is 10 mm/mV.

Echocardiography demonstrated global left ventricular hypertrophy with a dilated aortic root (Z-score calculated according to [7]: 2.5) and grade I aortic insufficiency (Figure 2).

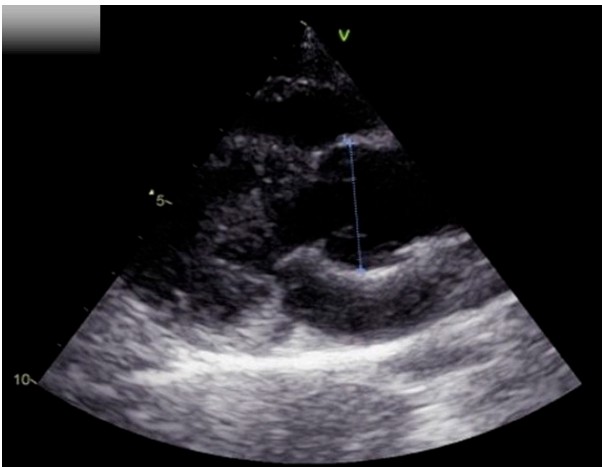

**Figure 2.** Echocardiographic modified long axis view with focus on the aortic root. The aortic root inner diameter in systole is 29 mm (Z-score: 2.5).

Abdominal ultrasound showed the presence of a right-sided adrenal mass (Figure 3).

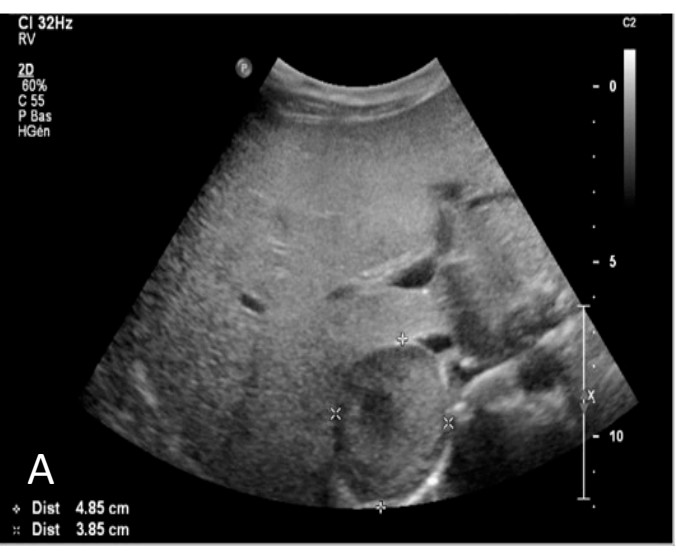
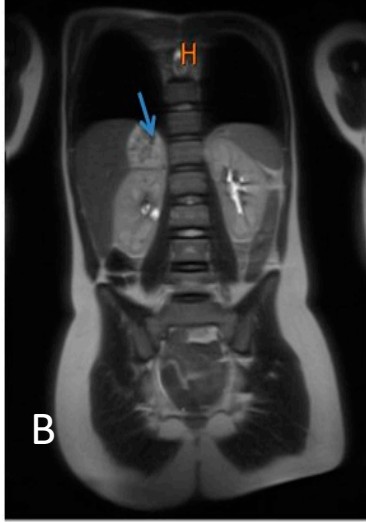

**Figure 3.** (**A**) Abdominal ultrasound demonstrating the presence of a right-sided adrenal mass of 4.85 × 3.85 cm; (**B**) the right-sided adrenal mass is shown at resonance magnetic imaging (arrow).

The left-sided adrenal gland and all other intra-abdominal organs were normal.

An oral glucose tolerance test showed moderate hyperglycemia associated with hyperinsulinemia. Cortisol- and thyroid hormone serum levels were normal. Noradrenaline and normetanephrine urine concentrations were strongly increased (>711 µg/24H and >9000 µg/24H, respectively). The right-sided adrenal mass was confirmed via magnetic resonance imaging (Figure 3), and iodine-123-metaiodobenzylguathdine ([123I]-mIBG) scintigraphy confirmed isolated tracer fixation on the right adrenal gland (Figure 4).

The patient was scheduled for surgery and stabilized until then using a combination of a β-blocker and calcium channel antagonist, allowing for blood pressure control. She underwent total adrenalectomy via laparoscopy. Anatomo-pathology confirmed the diagnosis of a typical pheochromocytoma without any signs of malignancy.

Microscopic analysis showed a nest-shaped tumor architecture with large polygonal cells. The central part of the tumor presented some ischemic necrotic areas and vascular structures with recent thrombosis but no signs of vascular invasion. There was no mitotic activity.

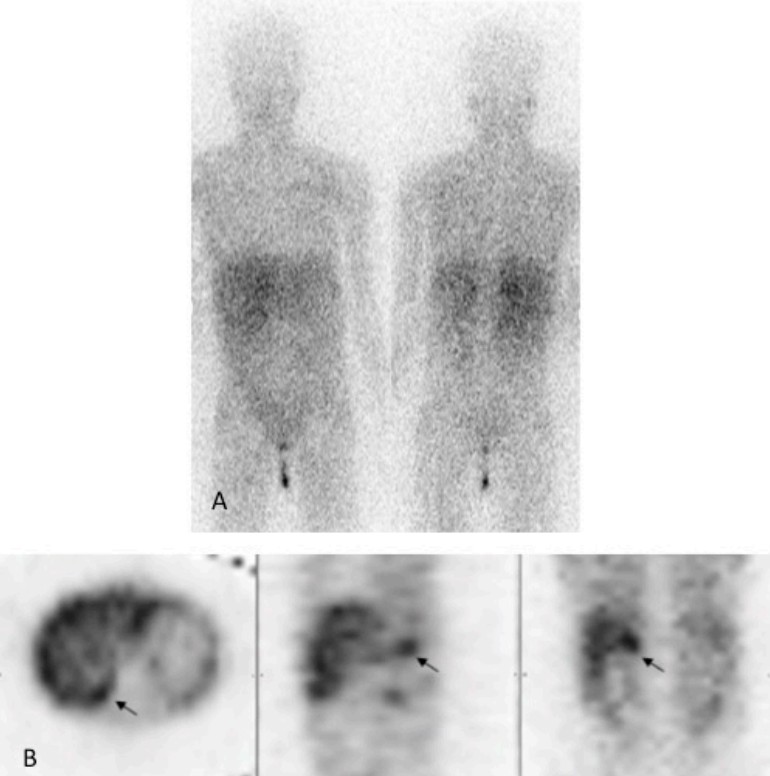

**Figure 4.** [$^{123}$I]-mIBG scintigraphy. (**A**) Anterior (left) and posterior (right) planar images show physiological uptake in the salivary glands and liver and urinary activity. (**B**) Axial, sagittal and coronal SPECT images demonstrate focal uptake (black arrow) above the superior pole of the right kidney corresponding to the adrenal lesion.

Immune-histological examination proved a high degree of positivity for the neural cell adhesion molecule CD56 and for the markers of neuroendocrine tumors chromogranine and synatophysine. The level of cell proliferation was estimated to be lower than 1%.

A genetic evaluation was carried out. DNA sequence analysis allowed for the exclusion of pathogenic gene mutations implicated in hereditary paragangliomas and pheochromocytoma (SDHA, SDHB, SDHC, SDHD, SDHAF2, TMEM 127, MAX, VHL, FH, MDH2 and exons 10, 11, 13, 14 and 16 of the RET gene) but demonstrated a somatic mutation c.57C>T in exon 3 of the tumor suppressor gene von Hippel-Lindau.

Cardiac examination, including echocardiography performed 2 weeks after right-sided adrenalectomy, showed residual left ventricular hypertrophy but normalization of the aortic root dimension (Z-score: 0.7) and QTc duration (Table 1).

**Table 1.** Comparison of ECG and echocardiographic parameters measured before and after the operation. The sum of S- and R-wave amplitudes measured respectively in chest leads V2 and V5 and the corresponding percentile value for age were used as diagnostic criteria of left ventricular hypertrophy according to [6]. QTc duration was assessed according to Bazzett's formula and the Z-score of the diameter of the aortic root according to [7].

|  | Pre-Operative | 2 Weeks Post-Operative |
| --- | --- | --- |
| SV2 + RV5 (percentile (P)) | 7.8 mvolts (>P98) | 6.1 mvolts (P95) |
| QTc duration | 470 ms | 304 ms |
| Aortic root size (Z-score) | 29 mm (2.5) | 25 mm (0.7) |

Two years after the operation, the patient complained about dizziness, palpitations and chest pain. The clinical examination, including blood pressure, was normal. Concentrations

of blood and urine catecholamines were in the normal range. Abdominal MRI showed a normal left-sided adrenal gland an empty right-sided adrenal lodge. Endocrinological and cardiological examinations, including echocardiography and abdominal MRI, were repeated and remain normal after a follow-up period of 5 years.

## 3. Discussion

Pheochromocytoma is a rare tumor arising from adreno-medullary chromaffin cells. Commonly, it manifests as headache, palpitations and hyperhydrosis in adults and in children as well [4,8].

In most patients, multiple cardiac manifestations are found, such as systemic arterial hypertension, a prolonged QT interval, dilated or hypertrophic cardiomyopathy, arrhythmias and more rarely myocarditis [9,10].

To the best of our knowledge, it is the first reported case of a child with pheochromocytoma and aortic root dilation. All yet reported cardiac manifestations of pheochromocytomas and paraganglioma are characterized by an acute and dramatic clinical presentation with a high complication rate, particularly in patients in whom diagnosis is delayed.

Indeed, the catecholamine storm induced by the tumor may provoke acute coronary vasospasm with myocardial ischemia, increased inotropism with a hypercontractile state of cardiomyocytes and possible sarcomere rupture. If the patient survives, myocardial cell necrosis will finally lead to the infiltration of inflammatory cells and myocardial fibrosis causing, in turn, ventricular dysfunction [11]. Systemic arterial hypertension is the most common cardiac repercussion of pheochromocytoma. It is the cardinal but not constant sign and is particularly suggestive when associated with the clinical triad consisting of headache, sweating and palpitations [12]. In children with systemic arterial hypertension, the prevalence of pheochromocytoma is 1.7% [10].

Although most patients have either sustained or paroxysmal systemic arterial hypertension, some patients present with malaise due to orthostatic hypotension as was the case in our patient. Indeed, the excess of catecholamine release may cause baroreceptor desensitization and consecutive hypotension [8].

Left ventricular hypertrophy is also a classic complication. A study of 26 consecutive patients with pheochromocytoma reported concentric left ventricular hypertrophy at echocardiography in 37.9% of them [9].

The cardiac manifestations of pheochromocytoma are the consequence of the catecholaminergic impregnation. The cardiovascular responses to released catecholamine depend mostly on their quality and quantity.

Indeed, epinephrine and norepinephrine have a predominant effect on beta-adrenergic receptors and increase inotropy and chronotropy. They also have, to a lesser extent, an alpha adrenergic effect that induces an increase in cardiac output and systemic vascular resistance [13].

It is the alpha adrenergic stimulation that is responsible for systemic arterial hypertension, while the beta effect rather leads to the hypertrophic response. However, due to the down-regulation of the adrenoreceptors as a consequence of prolonged excessive levels of circulating cathecolamines, these symptoms are not constant [14].

The prolongation of the corrected QT interval has been reported in 16% to 35% of patients with pheochromocytoma [15,16]. A prolonged QT interval favors the occurrence of malignant ventricular arrythmias, such as ventricular tachycardia, and even ventricular fibrillation, especially torsade de pointes responsible for syncopes or cardiac arrest [17]. The pathophysiology remains uncertain but may involve catecholaminergic discharge leading to repolarization abnormalities in particular; alpha-adrenergic stimulation could prolong the QT interval by lengthening the duration of the action potential [18]. However, the precise site of the repolarization alteration is not yet defined [17,19]. Insofar as our patient is concerned, we could not find any other cause for QT prolongation than the catecolaminergic impregnation due to the pheochromocytoma.

In children, aortic root dilation in the context of pheochromocytoma has not been reported previously. The mechanisms by which the aortic wall becomes distended in

cases of pheochromocytoma may be due, at least in part, to acute systolic systemic arterial hypertension. Beside this, aortic wall damage may also be involved. Indeed, cases of aorto-arteritis have been reported as complications of pheochromocytoma in young adults for whom symptoms lasted for months. The complex and multi-factorial mechanisms causing aortic wall inflammation involve mechanical stress due to systemic systolic hypertension and aortic wall ischemia due to vasa vasorum constriction leading in turn to media necrosis and elastic layer destruction [20]. A catecholamine triggered auto-immune disease has also been proposed as a possible cause of aortic wall damage since high titers of lupus erythematosus auto-antibodies were measured in patients with pheochromocytoma and normalized after tumor resection [21]. In adults, in addition to aortic wall injury, other vascular anomalies, such as aortic aneurysm and renal artery stenosis have been described in association with pheochromocytoma [22]. However, these kind of complications have not been reported in the pediatric population. In our patient, it remains unclear yet whether circulating catecholamines exerted a local inflammatory effect on aortic wall remodeling. As stated above, dimensions of the aortic root normalized together with the disappearance of the other cardiac manifestations soon after tumor excision, make inflammatory-mediated damage unlikely and support its acquired characteristics.

Our case highlights the fact that in a patient with alteration of the general state, cardiac manifestations, such as left ventricular hypertrophy, a prolonged QT interval and aortic root dilation, must suggest a diagnosis of a pheochromocytoma.

An unclear issue in our case is the particular family history of our patient. Indeed, there is currently no known link between pheochromocytoma and pericardial agenesia, both rare diseases presented by siblings. Our patient with pheochromocytoma showed a somatic mutation of the von Hippel-Lindau gene without an associated germline mutation. According to the literature, this mutation is extremely rare [23–25].

International guidelines recommend genetic testing be performed for all patients with pheochromocytoma regardless of their age and their personal or family history. Indeed, a constitutional mutation of a predisposition gene is present in more than 40% of patients. Since the risk of recurrence is estimated to be 1 per 100 person-years, close follow-up is recommended. This should be annual and include blood and urine concentrations of catecholamines as was performed for our patient.

Recommended follow-up should be lifelong, especially in patients carrying a higher risk of recurrence, such as those of younger age, patients with a paraganglioma, a germline mutation, with an extra-adrenal or large tumor, or with a syndromic disease [26,27].

Specific cardiological follow-up will depend on the type of complications induced by massive catecholamine production. While systemic arterial hypertension and electrophysiological anomalies usually resolve after tumor resection, cardiomyopathies due to long-lasting hyperactivation of the sympathetic nervous system or even acute coronary artery syndrome and consecutive myocardial ischemia determine long-term cardio-vascular morbidity [11].

## 4. Conclusions

Pheochromocytoma in children is a rare tumor, and its diagnosis might be challenging due to the non-specific and confusing clinical signs. Recognizing the cardiac manifestations as acquired electrophysiological and morphological alterations due to excessive catecholamine production may be lifesaving in accelerating diagnosis and offering prompt adequate curative treatment. Depending on the type of complications, cardiological examinations may be indicated in addition to the recommended long term oncological and endocrinological follow-up.

**Author Contributions:** Conceptualization, M.-C.S. and A.-S.P.; methodology, L.D., C.D. and M.-C.S.; software, C.D. and L.D.; validation, L.D., A.-S.P., R.H. and M.-C.S.; formal analysis, A.-S.P., R.H. and M.-C.S.; investigation, L.D., R.H. and M.-C.S.; resources, C.D. and M.-C.S.; data curation, M.-C.S.; writing—original draft preparation, L.D.; writing—review and editing, L.D. and M.-C.S.; visualization, A.-S.P.; supervision, R.H. and M.-C.S.; project administration, M.-C.S.; funding acquisition, none. All authors have read and agreed to the published version of the manuscript.

**Funding:** This research received no external funding.

**Institutional Review Board Statement:** Not applicable.

**Informed Consent Statement:** Not applicable.

**Data Availability Statement:** Not applicable.

**Conflicts of Interest:** The authors declare no conflict of interest.

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
