# Peer review of "Unusual Cardiac Manifestations of a Pheochromocytoma in a Girl"

_pediatrrep, doi:10.3390/pediatric15010019_

Round 1

Reviewer 1 Report

Unusual cardiac manifestations of a pheochromocytoma in a girl

This is a very interesting case report, which clearly shows that in our medical diagnostic process we should not omit some more rare diseases in children including phaeochromocytoma. Eventually, some clinical findings were at first not specific, and then it led to the ultimate finding of Von Hippel-Lindau gene mutation. However, the most spectacular determination is the link between the disease in a child and cardiological complication, that is, a dilated aortic root, not described before in medical reports, as authors have claimed.

Abstract-minor punctuation mistakes such as – unnecessary comma before “and”

“orthostatic dizziness, and abdominal pain for 4 weeks.”

Line 53-please exchange this a little awkward expression “She did not improve”

to a more professional one as for example “patients’ status” or something else.

Line 60-62

“Electrocardiogram showed left ventricular hypertrophy (LVH) with non-specific repolarization anomalies”

1.      In children there are different LVH markers in ECG than in adults, please, specify which ones were filled and add in in the description or in brackets, was it AV1+Rv6? Parameter or something else; is it additionally important for the reason that majority of physicians rather know criteria for adults only

Line 62 please change the misspelling of “msecondes” into milliseconds or ms

Line 63 it should be: “with a dilated aortic root”

Line 108 -there is no period mark “.”

Line 118-119

Please re -arrange this part: “To the best of our knowledge, our case is the first child to be reported to have additional aortic root dilation”

for example to” To the best of our knowledge, it is the first reported case of a child with pheochromocytoma and the dilated aortic root”.

144-145  “...of malignant ventricular arythlmias such as ventricular tachycardia or ventricular fibrillation/torsade de pointes reponsible for syncopes or cardiac arrest”

Please, correct small spelling mistakes, as well as re-arrange the sentence.

For example concluding: “…of malignant ventricular arrhythmia such as ventricular tachycardia, especially torsade de pointes, and even ventricular fibrillation…”

Author Response

Dear Reviewer,

thank you very much for the time spent to review our manuscript and for your comments.

All have been addressed as follows:

  1. This is a very interesting case report, which clearly shows that in our medical diagnostic process we should not omit some more rare diseases in children including phaeochromocytoma. Eventually, some clinical findings were at first not specific, and then it led to the ultimate finding of Von Hippel-Lindau gene mutation. However, the most spectacular determination is the link between the disease in a child and cardiological complication, that is, a dilated aortic root, not described before in medical reports, as authors have claimed.
    --> Thank you for your comment

2. Abstract-minor punctuation mistakes such as – unnecessary comma before “and”

“orthostatic dizziness, and abdominal pain for 4 weeks.”

--> fixed

3. Line 53-please exchange this a little awkward expression “She did not improve” to a more professional one as for example “patients’ status” or something else.
--> fixed

4. Line 60-62

“Electrocardiogram showed left ventricular hypertrophy (LVH) with non-specific repolarization anomalies”

  1. In children there are different LVH markers in ECG than in adults, please, specify which ones were filled and add in in the description or in brackets, was it AV1+Rv6? Parameter or something else; is it additionally important for the reason that majority of physicians rather know criteria for adults only

--> Thank you for this important point. Indeed, the classical Davignon's criteria were used to diagnose LVH. We now precise this and refer to the original publication giving the Gaussian distribution by age.

5. Line 62 please change the misspelling of “msecondes” into milliseconds or ms
--> Fixed

6. Line 63 it should be: “with a dilated aortic root”

--> Fixed

7. Line 108 -there is no period mark “.”

--> Fixed

8. Line 118-119

Please re -arrange this part: “To the best of our knowledge, our case is the first child to be reported to have additional aortic root dilation”

for example to” To the best of our knowledge, it is the first reported case of a child with pheochromocytoma and the dilated aortic root”.

--> Done

Thank you again for your comments and your help in improving this work.

Reviewer 2 Report

In "Case report" section, you only refer to a generic "surgery" without describing how the ablation of the tumour was performed: by robotic surgery,  by laparotomy or by laparoscopy? Total adrenalectomy or not? I think it's important to know.

Author Response

Dear Reviewer,

Thank you very much for your comments and the time spent to revise our manuscript.
Your comments have been addressed as follows:

1. In "Case report" section, you only refer to a generic "surgery" without describing how the ablation of the tumour was performed: by robotic surgery,  by laparotomy or by laparoscopy? Total adrenalectomy or not? I think it's important to know.
--> Thank you for this important point.
We precise now the type of surgery (total adrenalectomy by laparoscopy).

2. 144-145  “...of malignant ventricular arythlmias such as ventricular tachycardia or ventricular fibrillation/torsade de pointes reponsible for syncopes or cardiac arrest”

Please, correct small spelling mistakes, as well as re-arrange the sentence.

For example concluding: “…of malignant ventricular arrhythmia such as ventricular tachycardia, especially torsade de pointes, and even ventricular fibrillation…”

--> done

Thank you very much again for your comments and  your help in improving this work.

Reviewer 3 Report

Here d’Angelo et al. described a first-time interesting case of a girl with pheochromocytoma and an associated aortic root dilatation. This is a nice case report and well-written. I have just a minor notice. I think a table including the measurements in direct size and in z-Score for the aortic root as well as for the LVH would be nice to have. Otherwise, a really nice paper.

Author Response

Dear Reviewer,

Thank you very much for your comments and the time spent to review our manuscript.
Your comments have been addressed as follows:

  1. Here d’Angelo et al. described a first-time interesting case of a girl with pheochromocytoma and an associated aortic root dilatation. This is a nice case report and well-written. I have just a minor notice. I think a table including the measurements in direct size and in z-Score for the aortic root as well as for the LVH would be nice to have. Otherwise, a really nice paper.

    --> Thank you for your comments.
    A table has been added that shows the comparison of LVH signs, QTc duration and aortic root diameter before and after the operation.

    Thank you again for your comments allowing to improve this work.

Reviewer 4 Report

The clinical case presented by the Authors describes a peculiar occurance of pediatric pheochromocytoma in a patient characterized by rare genetic and clinical characteristics as well as familial agenesia of the pericardium. 
While the young patient did present a rather typical presentation of the disease, it is indeed interesting to note the presence of aortic root dilation and its resolution following surgical resection, a potential addition to telltale signs of pheochromocytoma. 

I find no particular flaws or points of criticism in the paper, apart from two extremely minors typos:

Line 61: "Bazett's formula", ditto for Figure 1's caption.
Line 62: "milliseconds"

Author Response

Dear Reviewer,

thank you very much for your comments and the time spend to review our manuscript.
Your comments have been addressed as follows:

1. The clinical case presented by the Authors describes a peculiar occurance of pediatric pheochromocytoma in a patient characterized by rare genetic and clinical characteristics as well as familial agenesia of the pericardium. 
While the young patient did present a rather typical presentation of the disease, it is indeed interesting to note the presence of aortic root dilation and its resolution following surgical resection, a potential addition to telltale signs of pheochromocytoma. 

I find no particular flaws or points of criticism in the paper, apart from two extremely minors typos:

Line 61: "Bazett's formula", ditto for Figure 1's caption.
Line 62: "milliseconds"

--> The mistakes have have been corrected.

Thank you again for your comments and the help to improve our work.